# Whole blood RNA extraction efficiency contributes to variability in RNA sequencing data sets

William W. Wilfinger[1]*, Hamid R. Eghbalnia[2], Karol Mackey[1], Robert Miller[3], Piotr Chomczynski[1]

1 Molecular Research Center, Inc. Cincinnati, OH, United States of America, 2 Department of Molecular Biology and Biophysics, UConn Health, Farmington, CT, United States of America, 3 Robert Miller Enterprises, LLC, Cincinnati, OH, United States of America

* billw@mrcgene.com

**Data Availability Statement:** • The original archived data files are available at NCBI Gene Expression Omnibus site (https://www.ncbi.nlm.nih.gov/geo/) containing sequencing data for the

## Abstract

Numerous methodologies are used for blood RNA extraction, and large quantitative differences in recovered RNA content are reported. We evaluated three archived data sets to determine how extraction methodologies might influence mRNA and lncRNA sequencing results. The total quantity of RNA recovered /ml of blood affects RNA sequencing by impacting the recovery of weakly expressed mRNA, and lncRNA transcripts. Transcript expression (TPM counts) plotted in relation to transcript size (base pairs, bp) revealed a 30% loss of short to midsized transcripts in some data sets. Quantitative recovery of RNA is of considerable importance, and it should be viewed more judiciously. Transcripts common to the three data sets were subsequently normalized and transcript mean TPM counts and TPM count coefficient of variation (CV) were plotted in relation to increasing transcript size. Regression analysis of mean TPM counts versus transcript size revealed negative slopes in two of the three data sets suggesting a reduction of TPM transcript counts with increasing transcript size. In the third data set, the regression slope line of mRNA transcript TPM counts approximates zero and TPM counts increased in proportion to transcript size over a range of 200 to 30,000 bp. Similarly, transcript TPM count CV values also were uniformly distributed over the range of transcript sizes. In the other data sets, the regression CV slopes increased in relation to transcript size. The recovery of weakly expressed and /or short to midsized mRNA and lncRNA transcripts varies with different RNA extraction methodologies thereby altering the fundamental sequencing relationship between transcript size and TPM counts. Our analysis identifies differences in RNA sequencing results that are dependent upon the quantity of total RNA recovery from whole blood. We propose that incomplete RNA extraction directly impacts the recovery of mRNA and lncRNA transcripts from human blood and speculate these differences contribute to the "batch" effects commonly identified between sequencing results from different archived data sets.

C9, C12 and C35 data sets in the following listings
GSE109484, GSE112057 and GSE169359.

**Funding:** The authors received no specific funding
for this work.

**Competing interests:** The authors declare that they
have no competing interests.

**Abbreviations:** mRNA, messenger RNA; lncRNA,
long noncoding RNA; GO, Gene Ontology; RNA-
seq, RNA sequencing; SD, Standard Deviation;
TPM, Transcripts Per Kilobase Million; RIN, RNA
Integrity Number; M, mean; CV, Coefficient of
Variation; DGE, differential gene expression.

# Introduction

Human blood has been recognized as an important diagnostic resource for centuries. Blood is
a complex fluid in continuous contact with all body tissues, thereby providing information
from a variety of unique compartments that include nucleated white blood cells (WBC), enu-
cleated red blood cells (RBC) and cell-free RNA: ribonucleoprotein complexes and ancillary
vesicular debris from assorted tissues in the body [1, 2]. Since blood collection is considered a
non-invasive procedure, it is frequently used for the evaluation of an assortment of disease
related biomarkers. In addition, the growing application of personalized medicine in the treat-
ment of chronic diseases has shown that RNA signatures can be employed to specifically opti-
mize treatment strategies best suited for the patient. This has supplied the impetus for the
development of a variety of unique methods for the collection, stabilization, and extraction of
RNA from blood.

In the clinic, blood is routinely collected in $K_2EDTA$ or ACD Vacutainer tubes, or with
blood collection tubes designed to stabilize RNA/DNA for subsequent extraction at some later
time (e.g., Pax Gene® [3–5, 6, 8, 9, 12, 14–18, 20, 21, 24–26], Tempus® [12, 14–17, 19–21,
23–26], RNAgard® [8, 9]). Each of the various blood stabilization tubes have unique propri-
etary ingredients designed to stabilize the nucleic acids. The extraction procedures routinely
employed to purify and recovery RNA from blood samples add additional variability since
they employ different extraction technologies such as: phenol-based extractions [3–11, 13, 22],
silica gel column purification procedures [3–6, 8–26], glass fiber extraction columns [26], mag-
netic bead extractions [16–17, 19, 23] and assorted blood cell enrichment methodologies cou-
pled with various extraction protocols [6, 8–11, 17]. The total quantity of RNA recovered from
whole blood differs significantly between these various extraction methodologies [3–26], but
the reported RNA purity ($A_{260/280} > 1.9$ and $A_{260/230}$ ratios $> 1.7$) and integrity-based RIN val-
ues (RIN $> 7$) are in the acceptable range for microarray and RNA sequencing studies [27, 28].

Although investigators employing these different extraction technologies use decent quality
RNA considered acceptable for RNA sequencing, when the sequenced transcripts from identi-
cal samples are compared, greater variation is observed between methods than across different
blood samples [6, 8–11, 14, 16, 21, 26]. Therefore, although substantial amounts of data have
been generated with these various blood collection and extraction methodologies, attempts to
pool the data sets for more comprehensive meta-analysis have had limited success. Several
reports evaluating different extraction methodologies concluded that RNA yield contributes
significantly to technical variation across methods [11, 12, 14, 25, 29]. We reported that RNA
content in human blood ranges from 6–22 μg / ml [7], reaching concentrations greater than
previously reported in the literature [3–6, 8–26].

Advancements in next generation sequencing (NGS) have significantly reduced the cost of
RNA analysis and expanded the interest in applying RNA sequencing to an array of disease
conditions. Current dogma relating to the suitability of RNA for sequencing applications, sim-
ply based on RNA purity and integrity, is inadequate. Other criteria are needed to reduce vari-
ability and improve agreement across various extraction platforms. To address this question,
we evaluated three archived data sets in which blood was collected and extracted with different
methodologies, but the resulting raw sequencing counts were processed under identical condi-
tions to minimize analytical pipeline induced variability [27, 28]. We evaluated a variety of
parameters such as the number of gene calls, transcript size distributions and call variance
with the goal of identifying factor(s) that might reduce sequencing variability and provide a
testable explanation for the large batch effects frequently reported when comparing similar
sequencing data sets [29]. Based on our analysis, we propose that the differential recovery of
short to midsized mRNA and lncRNA transcripts during RNA extraction directly affects the

character and breadth of the RNA library and its amplification, thereby disproportionally altering RNA calls over the entire range of transcripts. In addition, we demonstrate that the proportional relationship between transcript size and TPM counts, considered to be the fundamental requirement for RNA sequencing, is only attained when RNA is uniformly extracted and recovered across a complete range of transcript sizes. To address these shortcomings, we strongly recommend reporting RNA yield in all sequencing studies (e.g., μg RNA / ml of blood). We encourage investigators to select RNA extraction protocols that provide a mean RNA recovery approximating 14 μg RNA / ml of human whole blood [7] and that efficiently recovery RNA over the entire range of transcript sizes.

## Methods

### Ethics approval and consent to participate

The blood samples employed in this study were collected in accordance with the approved protocol provided by the Chesapeake Research Review, LLC. CIRBI Protocol # Pro00009509 [7]. Participants received a written informed consent form that was signed and witnessed in accordance with Chesapeake IRB guidelines. Participants provided witnessed signed informed consent documents specifically approved by Chesapeake IRB guidelines. All methods were performed in accordance with the relevant guidelines and regulations as outlined in the Declaration of Helsinki. The ethics requirements for the other archived data cited in this report are available in the public forum and were previously met when the data were originally published [33, 34].

### RNA extraction, sequencing, and data analysis

In an earlier report, we analyzed human whole blood RNA concentrations and differential blood cell counts in thirty-five individuals ranging in age from 50–89 years of age [7]. Fasting venous blood was collected with $K_2$EDTA Vacutainer tubes, stored at room temperature for about ~ 15 minutes and aliquoted into RNAzol-BD [7]. The blood: RNAzol-BD lysates were aggressively shaken to solubilize denatured proteins before storage at -70 C. Extraction of large RNA transcripts greater than 200 base pairs (bp) was performed according to the manufacture's protocol (https://www.mrcgene.com/product/rnazol-bd). The two hundred bp cutoff was established based on Bioanalyzer electropherogram plots of total, large and small RNA profiles [30]. The RNA was DNase-treated, and 1 μg of large RNA was sent to the University of Cincinnati Genomics, Epigenetics and Sequencing Core Facility for sequencing. After passing quality control analysis, the samples were depleted of globin and ribosomal transcripts prior to library formation and sequencing on the Illumina HiSeq 2000 platform (GSE169359). Standard procedures were employed to evaluate the quality of the raw data and the resulting FASTQ data files held 53–77 million single-end reads [31, 32].

We surveyed the Sequenced Reads Archives public repository (https://www.ncbi.nlm.nih.gov/sra) to select additional data sets for comparison. Archived data set one contained nine normal controls (GSE109313) ranging in age from 18–70 years of age was included in our analysis [33]. In this study, blood was collected with PAXgene collection tubes and then extracted with the PAXgene RNA extraction kit. A total of 500 ng of RNA was ribo-depleted and used for poly(A) selection. A second archived data set contained twelve control subjects (GSE112057) of unreported age [34]. Blood samples were collected in Tempus blood collection tubes and the RNA was extracted with the Tempus Spin RNA Isolation kit. The FASTQ data files held 21.5–49.3 million double-end reads.

These data sets represent diverse RNA extraction methodologies, providing an opportunity to examine how different extraction procedures might impact sequencing results. To minimize

data processing variability, all three data sets were processed through our pipeline under identical conditions, as outlined in an earlier report [31]. Briefly, FASTQ data files were trimmed and processed. Single-end reads were aligned to reference genome GRCh37.p13[hg19] using the BowTie2 aligner supporting gapped alignments. Cufflinks and HTSeq software were used to provide quality control for our analysis [31]. All counts were expressed as DeSeq-normalized TPM (Transcripts Per Kilobase Million).

A list of 25,354 sequenced read assignments was identified with the annotated reference genome. RNA transcripts containing < 0.1 TPM count were designated as 0 and RNA transcripts with means < 0.1 TPM counts were omitted from the analysis. The original source files were screened for copy number variants [32, 35, 36] and multiple assignments for individual genes were removed and expressed as a single gene ID. These data sets were used to compare and evaluate mRNA and lncRNA size distribution and transcript expression levels.

## Characterization of reference genome RNA transcript lengths

We examined the list of known RNA sequences from reference genome GRCH37.p13[hg19] to establish a baseline for comparison of transcript lengths in the three data sets [37]. This reference genome contained 258,705 entries ranging in size from 5 to 347,561 bp's sequences (S1 Table) with both mRNA and lncRNA sequences as well as known RNA sequences lacking a name or function (32,686 entries). The blue filled area in Fig 1 depicts the size and number of named transcripts assigned to each size interval (left y-axis). Each RNA transcript was assigned to an interval (bin, n = 200) based on $Log_{10}$ size range of 1 to 360,000 base pairs (e.g., $Log_{10}$ 0–5.556 bp using a 0.028 increment interval). To improve the clarity of the transcript size distribution plot in Fig 1, the gene $Log_{10}$ scale was limited to values of 1.7–4.4 thereby covering a size range of 50 to 30,000 bp's.

Transcript size does not follow a normal distribution and there is an obvious peak of 28,182 transcripts at ~550–566 bp. Most of the transcripts falling within this size range consist of lncRNA [38, 39]. In addition, larger transcripts (e.g., > 1,000 bp's) code for mRNA genes with multiple variants. To avoid averaging mRNA length measurements, transcript size is based on the size of variant one.

To further characterize the size distribution of the RNA reference genome depicted in Fig 1, we downloaded two additional files from NCBI containing annotated lists of mRNAs (82,961) and lncRNA transcripts (10,782) relevant to the same reference genome (right y-axis). Although the annotations for lncRNA transcripts relating to assigned function and accepted identification labels are not as mature as the assignments for mRNA transcripts, the size distribution of lncRNA is interspersed among the mRNA genes [38] and it may provide information relating to transcript recovery during RNA extraction. The three sequenced experimental data sets described in this study were evaluated in relation to the size distribution of the mRNA and lncRNA reference files depicted in Fig 1 (S1 Table of Transcript sizes).

In our analysis, single-pair sequencing transcripts were assigned to a single gene ID [31, 32] and specific variants were not identified. Therefore, variant one was selected to represent the gene length assignment of genes with multiple size variants. Based on the selection of variant one, a list of 19,608 NCBI mRNA transcripts was identified for our analysis (Fig 1, red line, right y-axis). These RNA transcripts range in size from 180 (ETDC) to 43,816 (MUC16) bp (S1 Table). Since less information is available relating to the predominant size of specific lncRNA's [38, 39], multiple size designations for specific lncRNA transcripts were averaged and a second list of 6,725 unique lncRNA transcripts was identified (Fig 1, green line, right y-axis). The established transcript size of the NCBI reference genome was used as the basis for characterizing the sequenced RNA in the three experimental data sets (S1 Table of Transcript Sizes).

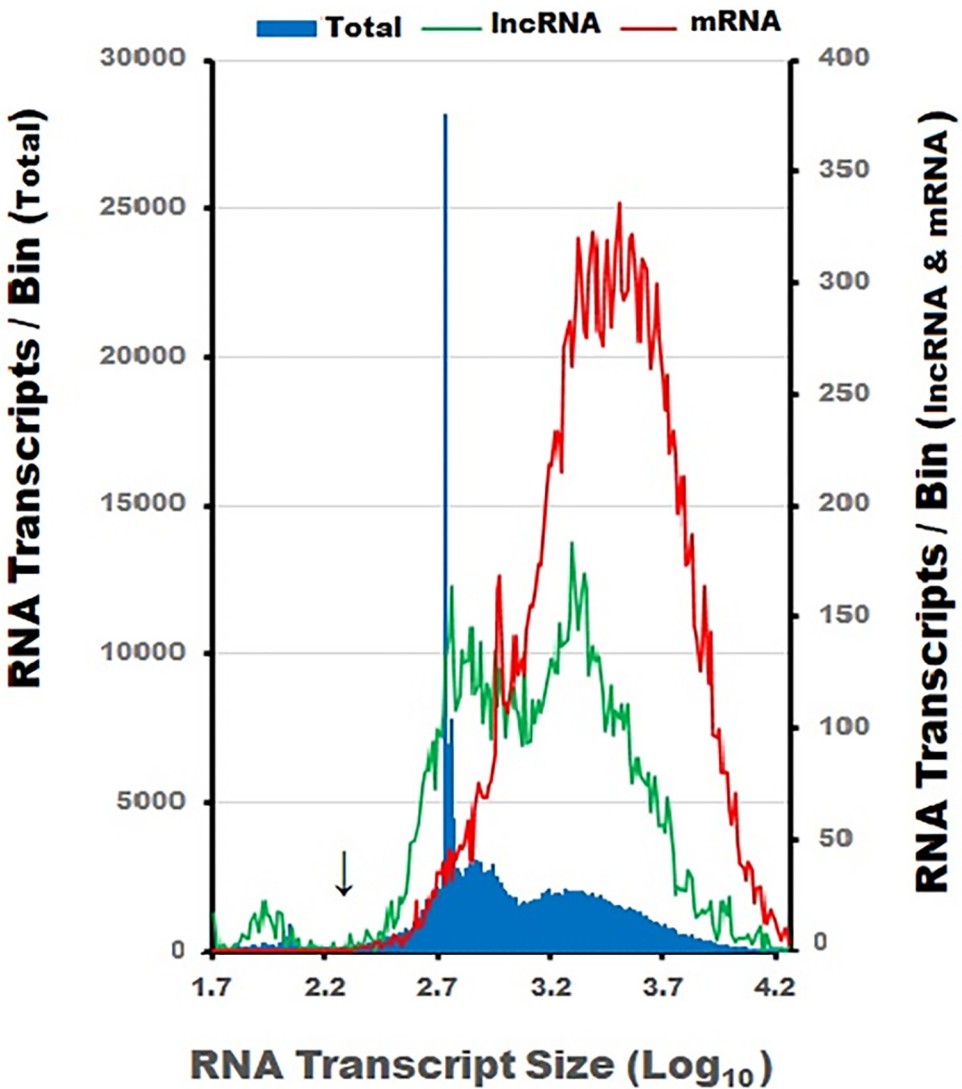

**Fig 1. The size distributions of NCBI transcripts.** The solid blue fill depicts the size of 226,019 RNA transcripts from reference genome GRCh37.p13[hg19]. RNA transcripts ranging in size from 5 to 347,561 bp were assigned to an interval (200 bins) based on their $Log_{10}$ base pair dimension (left y-axis). The x-axis scale range in Fig 1 is truncated and limited to transcripts ranging in size from 5–30,000 bp. The red line represents the size distribution of 19,608 mRNA transcripts (right y-axis) in which variant one represents the size designation of genes containing multiple size variants. The green line depicts the size distribution of 6,725 lncRNA transcripts. The black arrow identifies the 200 bp size cutoff for the RNAzol-BD extraction protocol [30]. A sizable proportion of the RNA depicted around 550–560 bp is associated with lncRNA transcripts [38]. The size dimensions of the mRNA and lncRNA transcripts used in our analysis overlap considerably.

## Characterization of RNA size distributions in experimental samples

The aligned reads from the three data sets were evaluated with SAMtools to quantify the number of reads per transcript [31]. Transcripts with mean raw counts $\geq 3$ were used to establish a baseline minimum count prior to TPM transcript size normalization. After TPM size adjustment, the source file containing a list of 25,354 transcripts was further evaluated. To provide the largest pool of transcript candidates, a preliminary list of transcripts was selected by limiting individual TPM counts to values $> 0.1$.

## Analytical methods

Statistical calculations were performed using the Microsoft Windows Excel Platform using the Analysis ToolPak application (Excel, version 2304 within Microsoft 365, Version 16.0. 16327). ToolPak is Charles Zaiontz's Real Statistics Resource Pack for Excel 2010, 2013, 2016, 2019, 2021 or 365 for Windows (Release 8.7). RAnGER data management software previously described in detail [32, 35] was employed to establish gene count minima, consolidate copy number variants [36], and perform standard statistical calculations. One-way ANOVA was employed in conjunction with Student-Newman-Keul's range test to evaluate computed means across the three data sets (S3 **Normalization and Statistical Analysis**).

## Results

### Overview of sequencing Results for RNA recovered from whole blood by different methodologies

The selected data files differ markedly in the way RNA was extracted and processed prior to RNA sequencing. We speculated that an examination and comparison of the sequencing results might indicate how these methods impact variability during RNA sequencing.

After processing the FASTQ data files, we looked at the total number of sequenced transcripts with mean TPM counts > 0.1 that corresponded to the NCBI reference genome. A summary of the analysis is presented in the Fig 2A table. The total number of identified transcripts ranged from 10,042 to 15,082 (33.4% difference) representing 55.4, 39.6 and 59.5% of the 25,354 annotated transcripts in the C9, C12 and C35 data sets, respectively. To further characterize these transcripts, mRNA and lncRNA transcripts with TPM Counts > 0.1 and known bp size assignments (Fig 1, S1 Table) were used to evaluate the original lists of total sequenced transcripts. Among the three data sets, mRNA and lncRNA transcripts with bp size assignments accounted for 84.8 and 6.70% of the identified transcripts, respectively. Collectively, total mRNA and lncRNA transcripts in the three data sets constitute 91.5% of the sequenced transcripts while transcripts with unassigned bp size assignments account for only 8.5% of the transcripts. Therefore, the transcripts with known bp size assignments should provide a representative assessment of the sequencing results. The RNA used for sequencing in the C9 and C35 data sets, on average, improved mRNA and lncRNA recovery relative to the C9 data set by 24 and 56%, respectively (S2).

To consider how RNA recovery affected reported transcript size, 19,608 mRNA and 6,725 lncRNA reference genome transcripts with known size assignments were used to characterize the sequenced mRNA and lncRNA transcripts identified in the data sets (S1 and S2). In the C35 data file, 11,882 mRNA transcripts and 1,195 lncRNA transcripts with known size measurement assignments were identified (Fig 2A). The 13,077 mRNA and lncRNA transcripts identified in the C35 data set was almost identical to the number of mRNA and lncRNA transcripts identified in the C9 data set (13,002) even though the two data sets were processed with markedly different methods (e.g., EDTA Vacutainer followed by phenol-based extraction with globin and mRNA depletion vs. PAXgene blood collection with silica gel column purification and poly-A selection). In contrast to the C9 and C35 data sets, fewer mRNA (24%) and lncRNA (56%) transcripts were identified in the C12 data set (Tempus blood collection with silica gel column purification of total RNA). The total number of transcripts identified in the C35 data set was 7 to 33% higher than in the other two data sets; however, a core group of 8721 mRNA transcripts was identified in all three data files and used in subsequent studies (Fig 2B).

If the number of mRNA transcripts identified in the C9 and C35 data sets is due to an improved recovery of RNA, one might expect to see more transcripts with TPM counts < 1 due to an improved recovery of low expression transcripts. Fig 2C clearly supports this

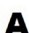

## A

| Data File | Total Transcripts Sequenced | mRNA Transcripts Identified | lncRNA Transcripts Identified | Total Identified mRNA and lncRNA Transcripts | Unassigned Transcripts |
|---|---|---|---|---|---|
| C9 | 14,051 | 11,981 | 1,021 | 13,002 | 1,049 |
| C12 | 10,042 | 9,069 | 494 | 9,563 | 484 |
| C35 | 15,082 | 11,882 | 1,195 | 13,077 | 2,005 |

**Summary of Identified Transcripts**

**B**

**Fig 2. Overview of transcripts in three sequenced data sets.** FASTQ-formatted data files from the three data sets were aligned and the mapped reads were assembled into transcripts. A) Tabular summary of identified sequenced RNA transcripts. The largest number of sequenced mRNA and lncRNA transcripts with TPM counts > 0.1 was found in the C35 data set, but the total number of transcripts with known bp size measurements was almost identical in the C9 and C35 data sets. In contrast, the number of mRNA and lncRNA transcripts was ~27% lower in the C12 data set (C35: 13,077 vs C12: 9,563 identified mRNA and lncRNA transcripts). B) Venn Plot analysis of the sequenced transcripts in the C9 (14,051), C12 (10,042) and C35 (15,082) data files identified 8721 mRNA and 481 lncRNA transcripts common to all three data sets. C) Size distributions of the total sequenced transcripts with mean TPM counts < 1. The number of sequenced transcripts with mean TPM counts < 1 is markedly higher in the C9 and C35 data sets implying an improved ability to detect weakly expressed transcripts.

inference. The total number of transcripts with TPM counts < 1 is greater within the C9 and C35 data set (C9 = 3413, C12 = 81 and C35 = 4409). Furthermore, the distribution of C9 and C35 transcripts presented in Fig 2C is shifted towards smaller transcripts as compared to C12. In summary, different numbers of sequenced mRNA and lncRNA transcripts were recovered with different extraction methodologies and in some cases, these differences are due to the improved recovery of weakly expressed short to midsized transcripts (Fig 2A and 2C).

### Size distribution of sequenced mRNA and lncRNA recovered with different extraction methodologies

To consider if a disproportionate recovery of RNA transcripts occurred during RNA extraction, the number of recovered transcripts was portrayed graphically in Fig 3, as a function of

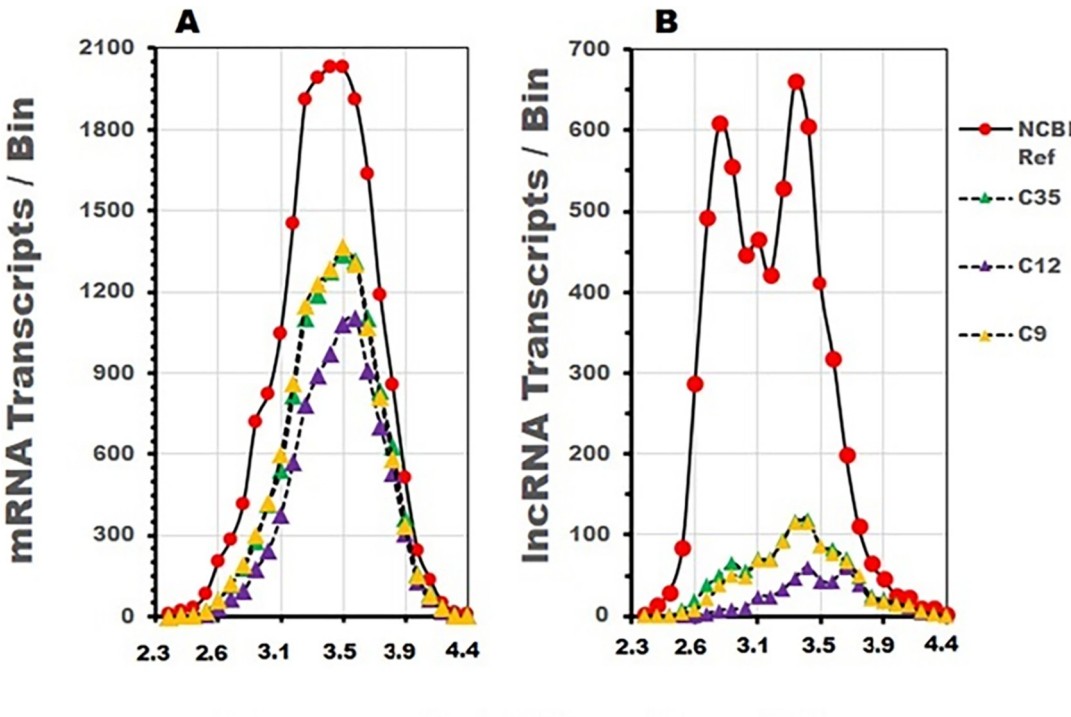

**Fig 3. Size distribution of mRNA and lncRNA sequenced transcripts extracted with different methodologies.** The size distribution of mRNA A) and lncRNA B) transcripts identified in the NCBI transcript reference list is plotted in relation to increasing transcript size measurements ranging from 200 to 30,000 bp (red circles) and used to provide a reference for sequenced RNA recovered from three independent data sets. The number of mRNA transcripts and their size distribution in the C9 and C35 data sets closely overlap while the C12 data set contains fewer total transcripts and a noticeable reduction of short and midsize transcripts. A smaller number of total lncRNA transcripts were identified in the C9 and C35 data sets and their distributions overlap. Similarly, to the mRNA size distributions in Fig 3A, the short and midsized lncRNA transcripts were also visibly diminished in the C12 data set.

transcript size. The size distribution of 19,608 mRNA genes (Fig 3A, red circles) and 6,725 lncRNA transcripts (Fig 3B, red circles) from the reference genome are presented with the size distributions of the transcripts recovered from the three data sets. The identified mRNAs and lncRNA for C9 (yellow), C12 (purple) and C35 (green) data sets were assigned a size designation and plotted relative to transcript size (Fig 3A and 3B). To improve clarity, only transcripts between 200 and 30,000 bp were reported. While the mRNA size plots for the C9 and C35 data sets overlap, the C12 transcript size distribution is smaller and shifted slightly to the right suggesting that short and midsized transcripts were omitted. When compared to the 2,752 smallest transcripts identified in the NCBI mRNA reference list that range in size from 200–1260 bp, the C9, C12 and C35 data sets contained 43.9, 24.7 and 42.4% of the reference transcripts, respectively.

The lncRNA transcripts between 200 and 30,000 bp identified in the NCBI reference list are depicted in Fig 3B (red circles). The biggest difference between the lncRNA reference list and the transcript profiles is the small number of identified lncRNA transcripts in the C9, C12 and C35 data sets, 15.2, 7.3, and 17.8% respectively. Like the mRNA profiles in Fig 3A, differences in the relative number of recovered lncRNA transcripts were most notable among the short and midsized transcripts in the three files.

## Analysis of 8,721 transcripts common among the three data sets

A foundational assumption in RNA sequencing predicts that the relative number transcript counts should be proportional to the size of the transcript. We used the 8,721 transcripts identified in all three data sets to see if this assumption held true for RNA extracted with the various methods.

All three data sets were adjusted to an equivalent number of total counts (200,000, S3) before preparing scatter graphs of the mean TPM counts plotted in relation to transcript lengths. Fig 4A–4C represents the normalized mean transcript expression levels for the three data sets plotted in relation to increasing transcript size over a range of 200 to 30,000 bp ($Log_{10}$ 2.3–4.4). After normalizing the three files to an identical number of total counts, the calculated mean for the 8,721 individual mRNA transcripts was 22.93 (solid red lines). The regression slope line for each distribution is depicted by the red-dashed line in each panel. The dashed red lines should have a slope of zero if the above assumption holds true.

The plots of the C9 and C12 data sets have negative regression slope lines of -1.1144 and -0.6469, respectively. The slope line of the C35 scatter plot approximates zero (-0.0643) and the distribution of mRNA transcript counts remains proportional to the relative size of the transcripts. The breadth of the scatter plot at any given transcript size range is explained by differences in the level of gene expression for transcripts of equivalent size. The three distribution plots represent an identical number of transcripts (8,721), so the computed slope lines could only be different from zero if TPM counts were disproportionate relative to transcript size. The C35 data set was the only file in which the number of TPM counts were proportional to the size of the transcript. It is also important to note that simply increasing the total number of sequenced transcripts in the C9 data set by poly-A selection did not correct or reduce the negative slope line of the mean transcript TPM count vs transcript size relationship. In fact, the negative slope line was 1.7-fold greater in the C9 data set than in the C12 data set. This result further emphasizes the importance of efficient transcript recovery on RNA sequencing outcomes.

## Coefficient of variation estimates of transcript variability

It is well known that variability, as estimated by the standard deviation (SD), increases in proportion to the calculated mean. To compare the variability among transcripts with different mean TPM counts, the coefficient of variation (CV) was employed. As previously noted, normalized TPM counts were used in these comparisons. To determine how this variability was affected by transcript size, scatter plots of the mean TPM count CV values for the C9, C12 and C35 data sets are presented in Fig 5A–5C. The distribution plot of the 8,721 transcript TPM count CV's is plotted as a function of transcript size ($Log_{10}$ bp). The solid red line (- 0.5 $Log_{10}$) represents a mean TPM count CV of ~ 0.31 or 31% for the 8721 transcripts. The scatter plot of the TPM count CV for the C9 and C12 data sets are similar and the positive regression slope lines suggest that the variability is increasing in relation to transcript size.

In contrast to the C9 and C12 data sets, the distribution plot of the mean TPM count CV values across the range of transcripts in the C35 data is uniform and decreases slightly, indicating the variability was greater among the smallest transcripts. This would be consistent with the premise that smaller transcripts yield fewer sequencing fragments, thereby providing fewer and more variable sequencing results [40, 41]. Finally, the narrow range of CV scatter across transcript size indicates that the C35 data file has the smallest overall level of variability as measured by the CV of 32.7% (C9 CV = 55% and C12 CV = 52%, respectively).

Based on our analysis of the 8721 transcripts common to all three data sets, using the mean TPM counts, and their variability presented in Figs 4 and 5, we speculated that the differences in the recovery of short and midsized transcripts contributes to these observed changes.

### Slope and intercept analysis of all transcripts in the three data sets

In our initial analysis of transcript TPM counts vs transcript size presented in Fig 4, total TPM counts were normalized across the three data sets and the regression analysis was limited to 8721 common transcripts. It is possible that the different slopes and intercepts observed in Fig 4, were influenced by the TPM count normalization or the selection of a reduced number of

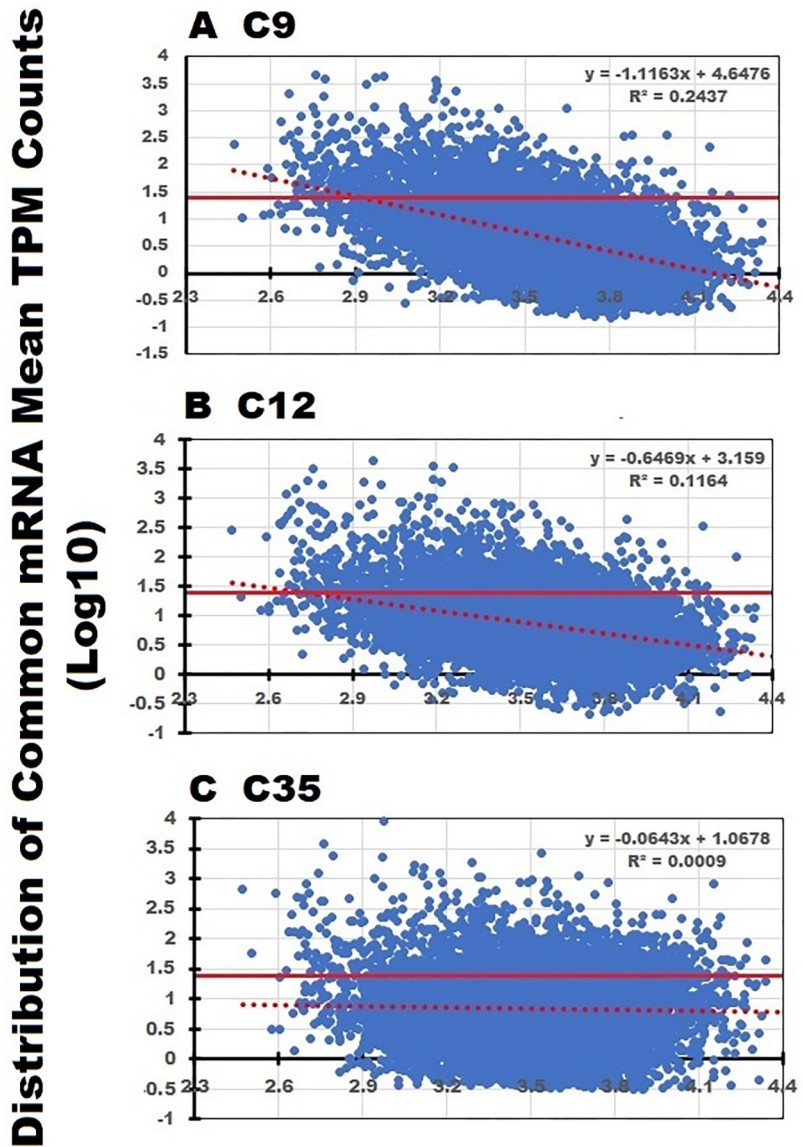

**Fig 4. Relationship of transcript mean TPM counts to transcript size among 8721 transcripts identified in all three data sets.** The mean TPM count for the individual transcripts was plotted in relation to the transcript size. The normalized sample mean of 22.93 is depicted by the solid red line. The mRNA transcript distribution plots depicted for C9 and C12 are similar, and both display negative regression lines (red dashed line). The solid and dashed red lines depicting the sample mean and regression line, respectively, of the C35 data sets are almost parallel. Since the total number of transcripts in all three data sets are identical, this result could only occur if the relative distribution of counts assigned to the distinct size transcripts has changed.

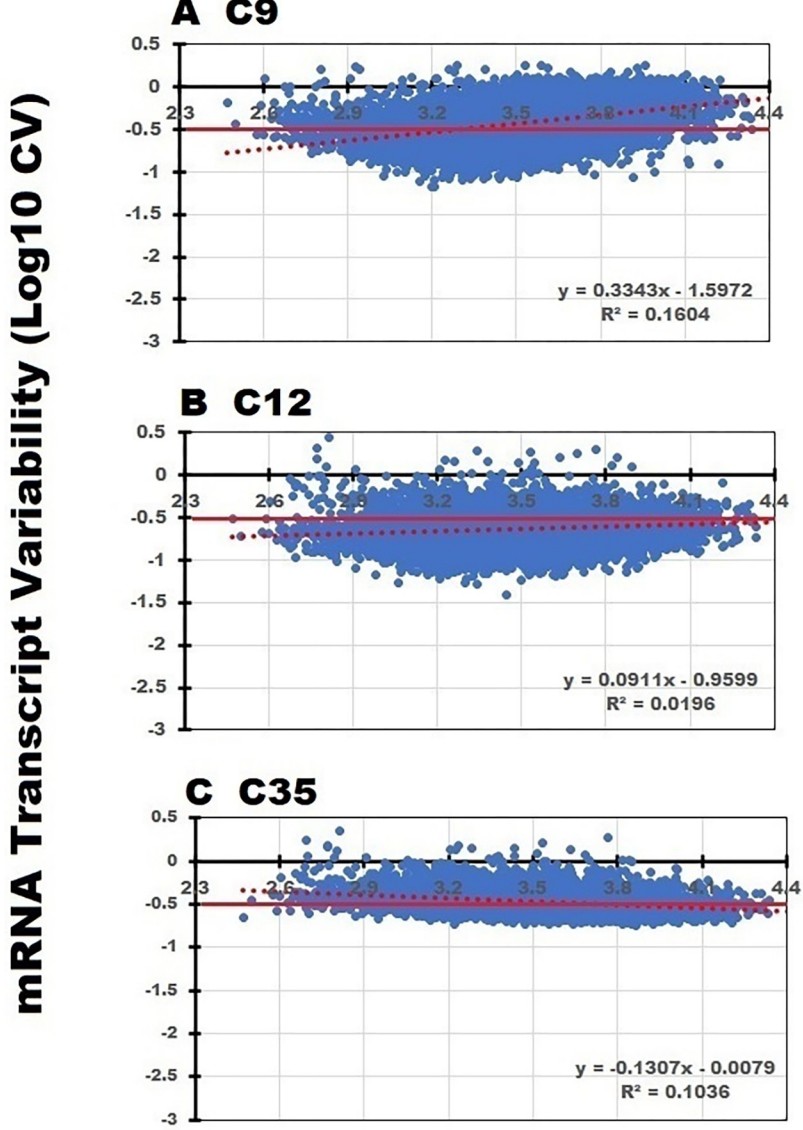

**Fig 5. Impact of transcript size on sequencing variability.** The positive slope line depicted by the red dashed line in the C9 and C12 data sets suggests that variability is increasing in relation to transcript size. Unlike the C9 and C12 data sets, the CV distribution within the C35 data set is less variable and more evenly distributed across all transcript sizes. The regression slope line of CV dispersion in the C35 data set is negative suggesting that the smaller mRNA transcripts are more variable, and the CV values decline with increasing transcript size.

total transcripts. To examine this possibility, we computed the individual slope and intercept values for the lncRNA and mRNA transcripts with known size measurements and TPM counts > 0.1 in all 56 samples. Since the individual samples have different numbers of total TPM counts and identified transcripts, regression analysis of the individual samples is likely to contain the greatest variability among the three data sets. The results of this comprehensive analysis are presented in the Box plots depicted in Fig 6.

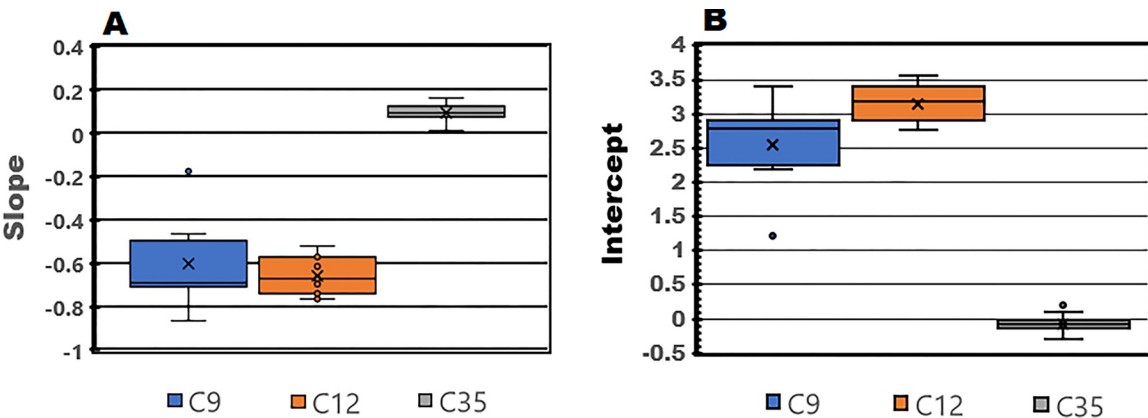

**Fig 6.** Box plots depicting the mean (x) slope (A) and intercept (B) values for individuals in the C9, C12, and C35 data sets. All identified mRNA and lncRNA transcripts with a TPM count > 0.1 were regressed in relation to transcript size. The average number of mRNA and lncRNA transcripts with TPM counts > 0.1 and known transcript size in the three data sets were 12,164 ± 500.7, 9108 ± 6.68 and 11,847 ± 261.1, respectively. The mean slope and intercept values recorded for the three data sets were proportional to values previously noted in Fig 4.

One Way Analysis of Variance of the computed slopes (C9: -0.600 ± 0.198, C12: -0.650 ± 0.083, and C35: 0.096 ± 0.038, Mean, SD) and intercepts (C9: 2.539 ± 0.620, C12: 3.142 ± 0.267, and C35: -0.083 ± 0.124, Mean, SD) of the regressed transcript-TPM counts vs transcript-size relationships in the three data sets clearly demonstrate that the slope and intercept of the C35 data set is markedly different than the values identified in the C9 and C12 data sets (df = 2, 53, slope: F = 411, P < 5.3E-33, Intercept: F = 711 P< 5.24E-39). Furthermore, the observed slopes and intercepts of the samples in the C35 data set are consistent with the expectations that form the basis for RNA sequencing analysis.

The colored area within each box plot represents approximately 50% of the values identified between the 25 and 75 percentile and the solid black line represents the sample median. In Fig 6, note the shift in the sample median from the mean in the C9 and C12 data sets relative to the C35 sample group and the degree of dispersion between the slopes and intercepts of the data sets. Regression analysis of the sequencing results of the individual samples without TPM count normalization or the survey of identical transcripts in the various data sets did not alter the unique slope and intercept profiles previously described in Fig 4. But the degree of sample dispersion is reduced in the C35 data set vs that noted for the C9 and C12 samples. A detailed summary of the statistical analysis is provided in S3.

## Discussion

A variety of methodologies are available for the extraction and recovery of RNA from whole blood. In addition, novel collection tubes, such as PAXgene and Tempus tubes, also can be employed for RNA stabilization [3, 4, 6, 8, 12–21, 23–26]. Although published microarray and RNA sequencing studies consistently use "good quality" RNA that is defined by A260/280, A260/230 ratios and RIN values, differences in gene expression are frequently reported and findings are difficult to replicate across various experimental platforms. The resulting differences observed between similar studies performed using different extraction methodologies preclude pooling the data for more potentially informative "meta" analysis, irrespective of the fact that these sequencing studies begin with "good quality" RNA. Apparently, extracted RNA is not identical across studies even though it has been deemed "good quality." Since the quality and integrity of the RNA are theoretically identical, other factors must be contributing to the

variability of the sequencing results. In this report, we compared three NCBI archived data sets of RNA sequencing results where markedly different extraction methods were used to obtain RNA from the blood of human control subjects. We sought to identify additional factors contributing to the disparate results commonly reported for sequencing results. Table 1 summarizes the factors addressed in this study.

In our initial survey of the sequencing results, we detected sizeable differences between the three data sets in the total number mRNA and lncRNA transcripts, as noted in Fig 2A. The number of sequenced transcripts in the C12 data file was ~28.5 and 33.7% lower than in either the C9 or C35 data sets even though the raw FASTQ files were processed identically. Furthermore, the identification of lncRNA transcripts in the C12 data set also was reduced by more than 50%. This is a substantial difference since total RNA was extracted for RNA sequencing in this data set. When evaluating RNA extraction, greater importance should be given to the quantity of RNA obtained per volume of blood (Table 1, A).

In our initial survey of mRNA and lncRNA transcripts, we identified major differences in the total number of transcripts as noted in Fig 2A. An examination of the number of transcripts with TPM counts < 1 among the three data sets also identified larger numbers of weakly expressed transcripts in the C9 and C35 data sets (Fig 2C). This finding implies that there is an improvement in the recovery of smaller and weakly expressed RNA transcripts in these two data sets. Therefore, the quantitative recovery of RNA is of considerable importance, and it should be viewed more judiciously (Table 1, B).

**Table 1. Summary of sequencing results recovered from data sets employing different extraction methodology.**

| Item | Experimental Parameter | Comment: |
|---|---|---|
| A | Extraction Capacity | The largest number of mRNA and lncRNA transcripts recovered from the FASTQ data files were found in the C35 data set (Fig 2A). Total transcript recovery was improved in relation to the C9 and C12 data sets by 6.8 and 33.4%, respectively). |
| B | Extraction Efficiency | The ability to efficiently recover short or weakly expressed transcripts independent of transcript size. The largest number of small and midsized transcripts with TPM counts < 1 was identified in the C35 data set (C9 = 3,413, C12 = 88 and C35 = 4,409, Fig 2A and 2C). |
| C | Recovery of small to midsized RNA transcripts | A comparative survey of mRNA and lncRNA transcripts based on transcript size revealed the differential loss of short to midsized transcripts in some data sets (Fig 3). |
| D | Proportional relationship of transcript counts to transcript size | The C35 data set was the only file in which the number of transcript counts could be shown to increase in parallel to transcript size, thereby fulfilling a fundamental precept of RNA sequencing (Fig 4C). |
| E | Experimental error is independent of transcript size | The coefficient of variation in the C35 data set was uniformly expressed over the entire range of transcript sizes (Fig 5C). In contrast, the CV increased in proportion to transcript size in the C9 and C12 data sets (Fig 5A and 5B), possibly due to incomplete transcript recovery. |
| F | Differential gene expression | If short to midsized mRNA and lncRNA transcripts are not efficiently and quantitatively extracted from whole blood, the increased variability attributed to transcript recovery across samples cannot be distinguished from differential gene expression. |
| G | RNA extraction | In additional to relying on the stability and purity of RNA recovered from whole blood, greater efforts should be directed at establishing criterion for evaluating the efficiency and capacity of the RNA extraction protocols that are routinely used for RNA sequencing. |

To gain additional perspective on the transcript size relationships of the known mRNA and lncRNA transcripts identified in the three data sets, the sequenced RNA transcripts were ranked according to their size measured in base pairs (bp's). The size distribution of sequenced mRNA and lncRNA transcripts recovered from the three data files is depicted in Fig 3A and 3B. The mRNA and lncRNA transcripts identified in the three data files are superimposed with the list of mRNAs and lncRNA transcripts from the NCBI reference files. A comparison of the mRNA and lncRNA transcript size distributions in the three data files, to the know transcript sizes of the NCBI reference genome, provides some perspective on the range of transcripts recovered during RNA extraction (Fig 3A and 3B). The visible absence of weakly expressed short to midsized mRNA and lncRNA transcripts in the C12 data set supports the conclusion that these transcripts may not be efficiently recovered in some extractions (Table 1, C).

Although major differences exist in the total number of identified transcripts between the three data files, 8721 common transcripts were identified in all three data sets (Fig 2B). When examining these transcripts, one would expect some degree of commonality since they represent blood drawn from healthy control subjects and they are expressed at sufficient levels to be detectable in all three data sets. When examined in relation to transcript size, the normalized mean TPM counts identified among the short to midsized transcripts in the C9 and C12 data sets appeared to be disproportionately higher than the mean TPM counts of largest transcripts. This unexpected result contradicts the long-held view that the number of sequencing calls should be proportional to the relative size of the transcript. However, the transcript count distributions in the C35 data set displayed proportional numbers of mean TPM counts across the entire transcript size range of 200 to 30,000 bp Fig 4C. We recommend confirming that the number of transcript counts remain proportional to the relative size of the sequenced transcripts (Table 1, D).

After observing differences in the slope lines of the transcript TPM count distributions across transcript lengths, it was of interest to see how these changes might impact data set variability. Since the CV is typically employed to evaluate the precision of a technique, ideally one would expect the CV to remain stable and independent of any changes in transcript size. Therefore, the coefficient of variation was employed to characterize transcript normalized TPM count mean variance among the three data sets. The CV plots for the C9 and C12 data sets (Fig 5A and 5B) are similar; however, the profile of the C35 data set displayed a much smaller range of dispersion over the entire transcript size range with a mean CV of 32.7% thereby approximating the inter-individual CV of 30.7% for total blood RNA concentrations [7]. The scatter plot slope line of the mRNA transcript means and their respective CVs, in the C35 data set, remained constant (slope $\cong 0$) over the entire range of transcript sizes from 200 to 30,000 bp's (Figs 4C and 5C). Therefore, when analyzing RNA sequencing data files, it may be useful to confirm that experimental error is evenly distributed and independent of transcript size (Table 1, E).

Many studies have been published to identify the factors that contribute to sequencing variability; however, our overall understanding has not progressed beyond the fact that RNA stabilization and extraction are fundamental sources of this variation [3–26]. Initial efforts to explain differences in the number of mapped reads identified protocol differences relating to the selection of RNA species [5, 27, 42]. rRNA and globin make up a substantial number of total RNA transcripts and if they are not removed prior to library formation, they will constitute most of the sequencing reads. Therefore, selection of total RNA versus poly-A selected RNA, or globin and rRNA depletion, have a major impact on library complexity and the number of mapped reads [27, 42]. The development of dependable and reproducible ribosomal and globin depletion protocols has significantly improved the number of exonic and intronic

reads that can now be detected during RNA sequencing [5, 14, 43–45]. Protocols that use total RNA extraction in conjunction with globin and ribosomal RNA-depletion procedures demonstrate significant improvement in the total number of mapped reads.

Although the mRNA and lncRNA transcript size distributions of the C9 and C35 data sets are identical (Fig 3A), the normalized transcript mean and CV distribution plots for these two data files are markedly different (Figs 4–6). While RNA transcript recovery is important, it is of equal importance to demonstrate that the sequenced transcripts exhibit a proportional transcript size/TPM count relationship as depicted in Fig 4C. Although similar numbers of total transcripts are recovered in both data sets, only the sequenced genes in the C35 data set show this proportionality. Apparently, the presence of a large number of transcripts is unable to correct deficiencies in RNA recovery and sustain the proportional transcript size/TPM count relationship observed in the C35 data set.

Based on the overview of the three data sets employed in this study, we speculate that the inefficient recovery of weakly expressed short to midsized RNA transcripts has a significant impact on RNA sequencing results. Sultan et.al. [5] previously reported an improved recovery of short RNA transcripts during phenol-based extractions and proposed that these transcripts are lost during complicated and tedious silica column-based extractions. Yip et.al. [21] has also reported that sample processing-dependent differences in gene expression were due to the loss of transcripts during RNA extraction. Therefore, it is reasonable to conclude that the loss of as many as 30–50% of the mRNA and lncRNA transcripts directly impacts the character and size of the resulting sequencing library. Sequencing results are further compromised during library amplification by a reduction in the total number of recovered transcripts as well as the incomplete and variable recovery of other transcripts resulting in over and under amplification of segments of the resulting library (Figs 4 and 5). If the primary character of the RNA library is skewed or misrepresented in any way, library amplification will further distort the sequencing results. Therefore, it is critically important to extract and efficiently recovery the complete range of transcripts from the blood during RNA extraction. The subsequent removal of globin and ribosomal RNA transcripts provides the best opportunity for the construction of a "complete" library that can be representatively amplified thereby significantly reducing the between file variance (Table 1, F).

We previously reported that the average RNA content in human whole blood is 14.58 μg / ml of blood with inter-individual variations ranging from 6.7 to 22.7 μg / ml (Inter-individual CV of 30.7% and intra-individual CV of 5.9%, [7]). Since the variation in inter-individual RNA recovery spans a 3.4-fold range, RNA extractions must have sufficient capacity to efficiently cover this dynamic range of blood RNA concentrations. Based on an average blood concentration of 14.58 μg of RNA / ml of blood, sequencing 1 μg of RNA from the C35 samples constitutes ~7% of the RNA in the sample. Previously reported column-based RNA yields for human blood are much lower ranging 1–8 μg of RNA / ml of blood [3, 4, 6, 8, 12–21, 23–26]. Using 1 μg of RNA from these extractions would constitute 12–100% of the RNA from the sample. The impact of these dramatic quantitative differences requires additional consideration, and we encourage investigators to pay greater attention to the amount of total RNA that is recovered during blood RNA extraction. Furthermore, we strongly recommend that the quantity of recovered RNA should be reported for every sample used for RNA sequencing and included in all publications. If the expected range of total RNA is routinely recovered and efficient globin and ribosomal RNA depletion protocols are employed there is a greater opportunity for a "highly complex" library to be identified in every sample and representatively amplified thereby significantly reducing sample variance [27, 42, 46].

In conclusion, we believe that differences in RNA recovery resulting from incomplete RNA extraction is a primary source of the RNA sequencing batch effects previously reported in the

literature. The disproportionate loss of short to midsize mRNA and lncRNA transcripts during RNA extraction excludes these transcripts from any subsequent downstream application such as qPCR, RNA microarray analysis or RNA sequencing. This issue may be extremely important when studying physiological pathways containing large numbers of short regulatory genes, such as cytokines, as well as the physiological role of short lncRNA transcripts [15, 16]. We speculate that these issues are further exacerbated when the cDNA libraries are amplified prior to qPCR or RNA sequencing. Finally, if the efficiency of RNA transcript recovery changes from sample-to sample with various extraction methodologies, it becomes impossible to determine the extent to which RNA recovery or the level of differential expression are contributing to the observed TPM count differences (Table 1, G). Although the methodological improvements mentioned here do not leap to the forefront when analyzing RNA sequencing data, addressing these concerns using the steps outlined here may uncover information buried in the data and strengthen study conclusions.

## Supporting information

**S1 File. Distribution of TPM counts across individual samples.** Maximum number of transcripts (N) identified in the C9, C12 and C35 data sets after processing FASTQ data files under identical conditions. A) A total of 12,059, 9,978 and 10,616 transcripts were found in every sample in the C9, C12 and C35 data sets, respectively. Further analysis identified meaningful numbers transcript counts as the number of samples per file was sequentially decreased to one. For example, in the C35 data set, 233 lncRNA transcripts were found in only 1 of the 35 samples but the counts ranged from 3.5–613 with a mean of 55.9 ± 85.9 TPM counts. Transcripts within this count range clearly represent legitimate gene expression values and thereby may provide some indication of the sensitivity of the extraction process.
(XLSX)

**S2 File. Normalization and statistical analysis.** Summary of the experimental rationale and ANOVA statistical analysis.
(DOCX)

**S1 Table. List of NCBI reference genome transcripts, mRNA and lncRNA transcripts with base pair size assignments.** To evaluate RNA recovery, transcripts in reference genome GRCH37.p13[hg19] [37] were used to identify the sequenced transcripts in the three data files. The lists of 19,608 mRNA and 6,725 lncRNA transcripts used to identify sequenced mRNA and lncRNA transcripts in the three data set are provided.
(XLSX)

## Acknowledgments

Robert Miller has furnished the programming for the RAnGER software that was employed in our analysis, and he has agreed to provide the use of the software in this publication. Copies of the software are available at rmillerllc927@gmail.com along with detailed video instructions for using the program [35].

## Author Contributions

**Conceptualization:** William W. Wilfinger, Karol Mackey, Piotr Chomczynski.

**Data curation:** William W. Wilfinger, Karol Mackey, Piotr Chomczynski.

**Formal analysis:** William W. Wilfinger, Hamid R. Eghbalnia.

**Funding acquisition:** Piotr Chomczynski.

**Investigation:** Karol Mackey.

**Methodology:** William W. Wilfinger, Hamid R. Eghbalnia, Karol Mackey.

**Software:** Robert Miller.

**Supervision:** Piotr Chomczynski.

**Validation:** Hamid R. Eghbalnia.

**Visualization:** William W. Wilfinger, Karol Mackey.

**Writing – original draft:** William W. Wilfinger.

**Writing – review & editing:** Hamid R. Eghbalnia, Karol Mackey, Robert Miller, Piotr Chomczynski.

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
