## [Decision Letter · Decision Letter 0]

26 May 2023

PONE-D-23-05931Whole Blood RNA Extraction Efficiency Contributes to Variability in RNA Sequencing Data SetsPLOS ONE

Dear Will,

I'm glad to report back that the manuscript looks good overall but requires minor updates. 

Please include more details of the methods as requested by both reviewers. 

Here are the Table formatting guidelines from the PLOS One website:

Tables

Cite tables in ascending numeric order upon first appearance in the manuscript file.

Place each table in your manuscript file directly after the paragraph in which it is first cited (read order). Do not submit your tables in separate files.

Tables require a label (e.g., “Table 1”) and brief descriptive title to be placed above the table. Place legends, footnotes, and other text below the table. 

Please add the table to the main paper text.

I look forward to receiving the updated manuscript soon. Once again, I apologize for the delays in the review process.

Cheers,

Surya

We look forward to receiving your revised manuscript.

Kind regards,

Surya Saha, PhD

Academic Editor

PLOS ONE

Journal Requirements:

3. Please amend your manuscript to include your abstract after the title page.

4. Please ensure that you include a title page within your main document. You should list all authors and all affiliations as per our author instructions and clearly indicate the corresponding author.

Additional Editor Comments:

Dear Will,

I'm glad to report back that the manuscript looks good overall but requires minor updates.

Please include more details of the methods as requested by both reviewers.

Here are the Table formatting guidelines from the PLOS One website:

Tables

Cite tables in ascending numeric order upon first appearance in the manuscript file.

Place each table in your manuscript file directly after the paragraph in which it is first cited (read order). Do not submit your tables in separate files.

Tables require a label (e.g., “Table 1”) and brief descriptive title to be placed above the table. Place legends, footnotes, and other text below the table.

Please add the table to the main paper text.

I look forward to receiving the updated manuscript soon. Once again, I apologize for the delays in the review process.

Cheers,

Surya

Reviewers' comments:

Reviewer's Responses to Questions

**Comments to the Author**

1. Is the manuscript technically sound, and do the data support the conclusions?

Reviewer #1: Yes

Reviewer #2: Partly

2. Has the statistical analysis been performed appropriately and rigorously? 

Reviewer #1: I Don't Know

Reviewer #2: I Don't Know

3. Have the authors made all data underlying the findings in their manuscript fully available?

Reviewer #1: Yes

Reviewer #2: Yes

4. Is the manuscript presented in an intelligible fashion and written in standard English?

Reviewer #1: Yes

Reviewer #2: Yes

5. Review Comments to the Author

Reviewer #1: Great study idea. I'd suggest in the future to have one data source (whole blood) with multiple aliquots so that your team can personally oversee the RNA extraction via different methods. There is likely more variability inherent within the sample sets than caused by extraction methods (ie: blood sample degradation, age of samples, environmental factors influencing gene expression).

In addition, find a way to include small non-coding RNAs in future assessments. Because of their short length, meaningful data can often be gained despite degraded blood samples. It would give a more comprehensive view: mRNA, lncRNA, and miRNAs paint a better picture than just mRNA or lncRNA.

I'd also suggest in future studies to outline your statistical methods in more detail. I understand the RAnGER was used for data management, but validation with a statistical software (even R, using the ranger package) would be a bonus.

Lastly, the figures need to be of higher quality. They are likely generated by a program or software (probably Ranger or similar) but find a way to export the figures in high quality or a larger size that you can then resize.

Reviewer #2: Table 1 is missing. This article cannot be accepted until it is included as it is referred to many times in the discussion section. While the article raises excellent points on the variability of RNA extraction and purification methods, it would have been stronger for them to perform all the RNA upstream steps themselves, as inter-lab variability is a common issue even for well established SOPs.

In the methods, it would be helpful to include the pipeline ran including versions, instead of referring to a previous paper. This applies to the extraction methods as well. A supplementary figure/table would suffice.

Overall the points the paper makes are valid. Including total RNA recovered would be useful for evaluating extraction procedures, with the understanding that any experimental samples could have largely different RNA amounts.

6. PLOS authors have the option to publish the peer review history of their article (what does this mean?). If published, this will include your full peer review and any attached files.

Reviewer #1: No

Reviewer #2: No

---

## [Author Response · Author response to Decision Letter 0]

19 Jun 2023

Response to Reviewers’

On behalf of my coauthors, I would like to thank the reviews for their constructive comments and editorial suggestions. To aid the reviewers in evaluating our responses, we reference specific text lines in the revised manuscript so the reviewers can quickly identify our edits. 

Reviewer 1:

General Comments to the author, Reviewers 1 and 2:

2. Has the statistical analysis been performed appropriately and rigorously? 

Both reviewers commented on this issue. We used the EXCEL Analysis ToolPak application (The ToolPak application employed in our study is part of Excel Version 2304 (Build 16327.20248), within Microsoft 365 version 16.0.16327) for our statistical analysis. ToolPak is Charles Zaiontz’s Real Statistics Resource Pack for Excel 2010, 2013, 2016, 2019, 2021 or 365 for Windows (Release 8.7). We have added this information in line 244 of the manuscript. In addition, in line 251 we reference supplemental file S3 as the location of more detailed information relating to our statistical analysis. 

We did not statistically evaluate the descriptive data summarized in Figure 2 since a comparison of the means of the three data sets would only provide an N of one. Our primary objective was to simply document that the three data sets contain different numbers of transcripts and that the recovery of small to midsized transcripts appears to be responsible for these differences (Figures 2 and 3). Previous publications (3-26) have reported significant differences in the total number of sequenced transcripts with various extraction protocols, but these observations have not resulted in the improvement of sequencing results or provided an explanation for the observed differences in differential gene expression between similar experimental data sets. 

We believe the principal contributions of this paper are the results presented in Figures 3, 4 and 5. Since the results presented in these figures were generated using the mean TPM counts of the various data sets, we followed the reviewers’ suggestions and performed a more rigorous statistical analysis. We have now calculated the individual slope and intercepts for all fifty-six samples used in this study to further quantitate the observations provided in Figures 3-5. The results of this additional analysis are presented in Figure 6 and the statistical analysis is described in supplement file (S3). We believe your suggestions strengthen and provide additional support for the important observations presented in Figures 4 and 5. 

General Comments to the author, Reviewer 1:

1. Prepare multiple aliquots of blood drawn from individuals and extract these samples with assorted RNA extraction methodology. We agree with your suggestion that this would provide useful information; however, this is a costly and time-consuming project that is unlikely to be covered by research grants or private funding. Therefore, a primary goal of our manuscript is to raise additional awareness to the fact that RNA recovery can be markedly different with various extraction kits and these differences have significant impact on RNA sequencing results. More importantly, we report that expressing transcript TPM counts in relation to transcript size is particularly useful in documenting how RNA extracts influence sequencing results independent of RNA purity and integrity, thereby providing a mechanism for evaluating the fundamental premises underlying RNA sequencing. We are hopeful that our manuscript will provide the rationale for the NIH to design and fund the project that you have suggested.

2. Include small non-coding RNA in future assessments. In human blood, small RNA constitutes approximately 23 % of the total RNA [7]. In our study, we elected to sequentially extract both the large and small RNA fractions in accordance with the RNAzol-BD extraction protocol. Based on our earlier report [7], using the large RNA fraction provides an additional 23 % enrichment of mRNA and lncRNA transcripts. The fact that small to midsized RNA transcripts were noticeably greater with the large RNA extraction protocol than with total RNA protocol employed with the other two methods further implicates the RNA recovery issue. Since we saved the small RNA fractions as you suggested, this could be the source of a future study. 

3. In future studies outline the source of your statistical methods. Thank you for alerting us to this omission. We used the EXCEL Analysis ToolPak for our statistical analysis. ToolPak is Charles Zaiontz’s Real Statistics Resource Pack for Excel 2010, 2013, 2016, 2019, 2021 or 365 for Windows (Release 8.7). This information is added to line 244 of the manuscript. In addition, more detailed information is given in supplement three and this is also referenced in line 251 of the revised manuscript. 

4. Rework the Figures to provide better clarity. We have downloaded and processed the Figures with the PLOS ONE PACE software and they were approved. If the academic editor considers that they may require additional modifications, we will comply with the directives of the journal. 

Specific Comments to the author, Reviewer 2:

1. Is the manuscript technically sound, and does the data support the conclusions? 

The response of Reviewer #2 to this question is “Partly”. To add to the weight of evidence, we have provided further quantitative data. The revised manuscript contains slope and intercept analysis presented in Figure 6 to provide stronger support for the conclusions presented in Figures 4 and 5. Please see my comments to reviewer # 1 for a more comprehensive explanation. 

Specific Comments Reviewer 2:

1. Table 1 is missing. Table 1 was originally sent to the journal as a separate file, and it appears to have not been included in the manuscript package sent to reviewer #2. The summary Table used in the Discussion section of the manuscript has now been incorporated into the revised manuscript. Please accept our apology for this omission! 

2. While the article raises excellent points on the variability of RNA extraction and purification methods, it would have been stronger for them to perform all the RNA upstream steps themselves, as inter-lab variability is a common issue even for well-established SOPs.

Based on the literature assessment of extraction efficiency, we would agree with your view that inter-lab variability is an issue. In a previous publication [7], we demonstrated that the intra-individual Coefficient of variation (CV) for our samples was 5.87 % while inter-individual variation was 5.2-fold larger (30.7%). Our samples were extracted by three different individuals over a period of approximately 1 month; however, the variability observed in data sets C9 and C12 is still markedly higher than in the C35 sample group as illustrated in Figures 5 and 6. We believe the variability attributed to inter-lab variability is due to incomplete RNA recovery during RNA extraction as well as during the column purification steps. We addressed this issue in our comments to reviewer one.

3. In the methods, it would be helpful to include the pipeline ran including versions, instead of referring to a previous paper. We have added the additional information relating to our data processing pipeline in the paragraph beginning at line 163 in the revised manuscript. Also note line number 223-229 in the original manuscript.

I have also provided an excerpt from one of our previous reports [31] to provide additional information for the reviewers.

Workflow. Raw data from thirty-five samples in the FASTQ format was submitted for quality control to evaluate adapter contamination, average base quality score per read, the GC content distribution, and other relevant parameters. The software FastQC was used to perform quality control. All samples received passing marks according to FASTQC benchmark criteria. FastQC software was further utilized to verify the quality of aligned BAM files. The alignments were generated by the Bowtie2 aligner, a BWT-based unspliced aligner, with the recent addition of supported gapped alignments. Bowtie2 extracts multiple substrings or seeds from each read and aligns them using a BWT approach with no gaps, then extends alignments using a Smith-Waterman-like scoring scheme. We used the results from Cufflinks for comparative analysis with the unique counts we extracted using HT-Seq and our in-house scripts. The GRCh37.p13 [hg19] reference genome was used in this study.

4. Overall, the points the paper makes are valid. Including total RNA recovered would be useful for evaluating extraction procedures, with the understanding that any experimental samples could have different RNA amounts. We agree that individual samples have different RNA amounts, and this information was reported for our thirty-five samples in our initial report (S1 Table, Ref 7), but sample RNA recovery was not reported with the sequencing files of the other two data sets. We also considered comparing the total number of identified mRNA and lncRNA transcripts among the three data sets. In Figure 2, we demonstrate that the total number of recovered transcripts with known size measurements in the C9 and C35 data sets are identical (C9: 13,002 vs C35: 13,077 transcripts); however, the data comparisons in Figures 3, 4 and 5 illustrate major differences between the two data sets. Although a similar number of transcripts were recovered in both data sets, the presumed incomplete recovery of small to midsized transcripts in the C9 data set accounts for the differences observed between the two data sets in Figures 3-6. Therefore, we believe a most crucial point presented in our paper is that the RNA extracted for RNA-seq applications must adhere to the basic principles that form the foundation of RNA sequencing analysis as illustrated in Figures 4 and 6.

---

## [Decision Letter · Decision Letter 1]

24 Aug 2023

Whole Blood RNA Extraction Efficiency Contributes to Variability in RNA Sequencing Data Sets

PONE-D-23-05931R1

Dear Dr. Wilfinger,

Congratulations!

We’re pleased to inform you that your manuscript has been judged scientifically suitable for publication and will be formally accepted for publication once it meets all outstanding technical requirements.

Kind regards,

Surya Saha, PhD

Academic Editor

PLOS ONE

Additional Editor Comments (optional):

Reviewers' comments:

Reviewer's Responses to Questions

**Comments to the Author**

1. If the authors have adequately addressed your comments raised in a previous round of review and you feel that this manuscript is now acceptable for publication, you may indicate that here to bypass the “Comments to the Author” section, enter your conflict of interest statement in the “Confidential to Editor” section, and submit your "Accept" recommendation.

Reviewer #1: All comments have been addressed

Reviewer #2: All comments have been addressed

2. Is the manuscript technically sound, and do the data support the conclusions?

Reviewer #1: Yes

Reviewer #2: Yes

3. Has the statistical analysis been performed appropriately and rigorously? 

Reviewer #1: Yes

Reviewer #2: Yes

4. Have the authors made all data underlying the findings in their manuscript fully available?

Reviewer #1: Yes

Reviewer #2: Yes

5. Is the manuscript presented in an intelligible fashion and written in standard English?

Reviewer #1: Yes

Reviewer #2: Yes

6. Review Comments to the Author

Reviewer #1: Thanks for addressing the concerns from the first review. This is an interesting study and the results are well written.

Reviewer #2: The authors have addressed my comments adequately and thoroughly. I recommend that they continue to push for better standards in the RNA field so that it's easier to compare between studies and datasets.

7. PLOS authors have the option to publish the peer review history of their article (what does this mean?). If published, this will include your full peer review and any attached files.

Reviewer #1: No

Reviewer #2: No

---

## [Editor Report · Acceptance letter]

29 Aug 2023

PONE-D-23-05931R1 

Whole Blood RNA Extraction Efficiency Contributes to Variability in RNA Sequencing Data Sets  

Dear Dr. Wilfinger:

I'm pleased to inform you that your manuscript has been deemed suitable for publication in PLOS ONE. Congratulations! Your manuscript is now with our production department. 

Kind regards, 

on behalf of

Dr. Surya Saha 

Academic Editor

PLOS ONE